# Updated Perspectives on Direct Vascular Cellular Reprogramming and Their Potential Applications in Tissue Engineered Vascular Grafts

**DOI:** 10.3390/jfb14010021

**Published:** 2022-12-30

**Authors:** Saneth Gavishka Sellahewa, Jojo Yijiao Li, Qingzhong Xiao

**Affiliations:** 1William Harvey Research Institute, Faculty of Medicine and Dentistry, Queen Mary University of London, London EC1M 6BQ, UK; 2Key Laboratory of Cardiovascular Diseases, School of Basic Medical Sciences, Guangzhou Institute of Cardiovascular Disease, The Second Affiliated Hospital, Guangzhou Medical University, Guangzhou 511436, China

**Keywords:** cellular reprogramming, cell transdifferentiation, direct cellular lineage-conversion, tissue engineered vascular grafts, vascular regeneration, stem cells, vascular progenitor cells, smooth muscle cells, endothelial cells, atherosclerosis, cardiovascular disease

## Abstract

Cardiovascular disease is a globally prevalent disease with far-reaching medical and socio-economic consequences. Although improvements in treatment pathways and revascularisation therapies have slowed disease progression, contemporary management fails to modulate the underlying atherosclerotic process and sustainably replace damaged arterial tissue. Direct cellular reprogramming is a rapidly evolving and innovative tissue regenerative approach that holds promise to restore functional vasculature and restore blood perfusion. The approach utilises cell plasticity to directly convert somatic cells to another cell fate without a pluripotent stage. In this narrative literature review, we comprehensively analyse and compare direct reprogramming protocols to generate endothelial cells, vascular smooth muscle cells and vascular progenitors. Specifically, we carefully examine the reprogramming factors, their molecular mechanisms, conversion efficacies and therapeutic benefits for each induced vascular cell. Attention is given to the application of these novel approaches with tissue engineered vascular grafts as a therapeutic and disease-modelling platform for cardiovascular diseases. We conclude with a discussion on the ethics of direct reprogramming, its current challenges, and future perspectives.

## 1. Introduction

Cardiovascular disease (CVD) is an increasingly prevalent cause of global morbidity and mortality and affects half a billion people worldwide [1]. Endovascular and surgical revascularisation therapies have been increasingly applied in patients with severe CVD. However, such measures are still limited and fail to recapitulate healthy arterial tissue, as evidenced by that almost 1 in 2 autologous saphenous vein grafts will experience graft failure by 10 years post-surgery due to factors such as vulnerability to arterial pressure, intimal hyperplasia, and continued atherosclerosis [2,3]. Moreover, small diameter vessels (<6 mm) commonly found in cerebral, cardiac and peripheral regions are hard to treat with limited grafts, poor surgical accessibility and subpar performance of synthetic polymer prosthetics [4].

To overcome the abovementioned limitations and present a new treatment modality, research has looked to an innovative regenerative approach known as direct reprogramming (synonyms include transdifferentiation and direct lineage-conversion) defined as “the process of inducing a desired cell fate, by converting somatic cells from one lineage to another without transitioning through an intermediate pluripotent or multipotent state” [5]. Direct reprogramming is a descendent of a large body of cellular reprogramming research which, alongside the Nobel Prize-winning work by Takahashi and Yamanaka, has shown that cell fate is not locked but can be manipulated through the ectopic expression of pluripotency factors, lineage-specific transcription factors, small molecules and non-coding RNAs [6]. The key galvanizing feature of this approach is the avoidance of pluripotency or multipotency. In contrast, induced-pluripotent stem cell (iPSC) generation requires the dedifferentiation of somatic cells to a pluripotent cell which then differentiates to the required lineage, often requiring ex vivo expansion [5]. Consequently, iPSC generation comes with multiple distinct limitations including costly, time-consuming, complex cell generation protocols, and tight regularity oversight due to the risks of tumorigenicity [7].

Multiple reprogramming approaches have emerged to generate vascular cells from somatic cells. The first protocol is using a partial reprogramming approach to generate partially iPSCs (PiPSCs). Specifically, the four pluripotency factors (OSKM; OCT3/4, SOX2, KLF4 and C-MYC) discovered by Takahashi and Yamanaka [6] is initially transfected into somatic cells for a short period (normally up to seven days) to produce PiPSCs. When placed in differentiation mediums, the PiPSCs further differentiate to the desired cellular lineage. Another common protocol uses a single or combination of lineage-specific transcription factors to induce the expression of lineage-specific genes while silencing original somatic cell genes by modulating chromatin configurations of target genes [5]. Two other approaches use small molecules to either induce a progenitor-like state [8] or activate innate immunity to form a state of epigenetic plasticity [9] which is sensitive to differentiation signals.

To provide an in-depth knowledge of the current arterial direct reprogramming landscape and future research directions, we conducted the narrative literature review by identifying literature sources from peer-reviewed journals on the PubMed and MEDLINE databases in the last 10 years (2012–2022). The key search terms used in this review included: cellular reprogramming, direct cellular reprogramming, transdifferentiation, cellular lineage-conversion, somatic cells, vascular cells, endothelial cells, smooth muscle cells, vascular progenitor cells, vascular regeneration and tissue engineered vascular grafts. The search terms were combined in various combinations with the ‘AND’ command to identify primary literature sources that specifically explored direct reprogramming approaches and their application in TEVGs. Any sources solely exploring iPSCs reprogramming approaches were excluded. Particularly, in this review we examine the generation of each arterial cell group (endothelial (ECs), smooth muscle (SMCs), and vascular progenitor (VPCs) cells) with an analysis of the specific reprogramming factors and their molecular pathways. We also briefly discuss how direct reprogramming strategies could be applied to tissue engineered vascular grafts (TEVG) for disease-modelling and for clinical application as vascular conduits. Finally, we conclude the review with a discussion on the ethics of direct reprogramming, its current challenges, and future perspectives.

## 2. Endothelial Cell Generation

Endothelial cells are found as a continuous monolayer in the tunica intima of arteries and directly interface with the bloodstream. Their roles are numerous and include the regulation of haemostasis, vascular tone, immunity, and angiogenesis. Here, we explore several strategies to directly generate functional ECs: pluripotency factors-, lineage-specific transcription factors-, innate-immune activation-, and microRNA-based reprogramming (Table 1; Figure 1).

### 2.1. Pluripotency Factor-Based Reprogramming

Direct reprogramming with the Yamanaka factors to generate induced-endothelial cells (iECs) was first demonstrated by Margariti et al. [10]. 4-day lentiviral overexpression of OSKM successfully dedifferentiated human fibroblasts into a partially induced-pluripotent state which did not express pluripotency markers such as SSEA-1 and did not generate any tumours when injected into mice. With the addition of endothelial differentiation culture medium (EGM-2 media), iECs were formed displaying typical endothelial morphology and functions (Low-density lipoprotein (LDL) uptake and angiogenesis), as well as expressing a panel of endothelial-specific markers such as vascular endothelial growth factor receptor 2 (VEGFR2; also known as kinase domain receptor, KDR), endothelial nitrous oxide synthase (eNOS), and von Willebrand factor (vWF). eNOS produces nitric oxide (NO), which vasodilates arteries, controls cell growth and resists inflammatory changes such as platelet aggregation. NO production is an important indicator of endothelial function and exerts an atheroprotective effect. However, this study did not assess NO production [10].

Based on Margariti and colleagues’ discoveries, it was soon shown that not all Yamanaka factors are required for transdifferentiation. Li et al. showed that OCT4 and KLF4 are sufficient for successfully reprogramming human fibroblasts into iECs, albeit with lower conversion efficacies of around 1%, improved to 3.85% with 8-Br-cAMP [11]. This conversion efficacy is significantly lower when compared to the 3-factor combinations of OCT4, SOX2 and KLF4 at 11.8% [11] and 4-factor combinations of OSKM at ~30% [10,12]. Nevertheless, sorting methods for the two-factor protocol achieved a 97% pure population of cells expressing a key endothelial marker, platelet endothelial adhesion molecule-1 (PECAM-1; also known as cluster of differentiation 31 or CD31) [11]. Using fewer factors may achieve cheaper and faster protocols while avoiding oncogenic factors like C-MYC. As technology improves and more specific markers are identified, selection and sorting methods may overcome low transdifferentiation efficacies. Indeed, such methods will need to be robust as any potentially pluripotent cell that does escape filtering can exert a teratoma risk [20].

Hong et al. showed that arterial SMCs were also amenable to the Yamanaka factor-based reprogramming approach [12]. SMCs are a valuable cell source as they are abundant and found immediately adjacent to the endothelium. Moreover, they possess significant phenotypic plasticity and share mesodermal origins and common progenitors with ECsduring embryogenesis, suggesting that their genetic and epigenetic mountains are easier to climb in the transdifferentiation process [21]. Hong and colleagues first generated CD34-positive vascular progenitors with 4 days of OSKM overexpression which was followed by differentiation in EGM-2 media and VEGF for 6 days [12]. Within their population of iECs, 33.4% of cells expressed CD31. Other key endothelial genes were upregulated (CD34, KDR, CD144, eNOS and vWF) while SMC genes were downregulated (α-SMA, SM22α, calponin, SM-MHC). Representative of functional mature ECs, the iECs took up LDL and increased expression of an inflammatory molecule, intercellular adhesion molecule 1 (ICAM-1) in response to TNF-alpha stimulation [12].

Hong and colleagues also identified several key insights into the reprogramming mechanism. The mesenchymal-to-epithelial transition (MET) occurred simultaneously with the reprogramming of SMCs to a vascular progenitor [12]. The upregulation of vascular endothelial cadherin (VE-cadherin, also known as CD144), an essential endothelial homeostasis and angiogenesis regulator, was the key event in the transition. In addition, two components within the Notch pathway, HES5 (HES family transcription factor 5) and JAG1 (Jagged canonical Notch ligand 1, a surface protein), had essential regulatory functions in differentiation. Overexpression of both proteins increased EC marker expression, while knockout studies on HES5 lowered marker expression. JAG1 increased the promoter activity of HES5 and eNOS. The Notch signalling pathway is affiliated with various cellular functions from cell growth, cell fate regulation and angiogenesis [22]. In keeping with the above findings, a recent pioneering study confirmed Notch signalling as a promoter of MET by increasing JAG1 mRNA expression through the RBP-Jκ transcription factor [23]. Moreover, JAG1 expression by ECs has been shown to propagate the development of multi-layered SMCs around an endothelial layer through lateral induction, which holds an exciting research avenue where JAG1/Notch signalling can facilitate reconstitution of an arterial wall from induced cells [24]. Furthermore, crosstalk between Notch and VEGF has been shown to promote angiogenic sprouting, vascular branching and stabilise cell-to-cell junctions [25].

Although the above studies have demonstrated no in vivo tumour formation for the respective observation periods in mice, the potential for tumorigenesis cannot be ruled out with short-term overexpression of pluripotency factors. Lentiviral delivery of the factors is rather troublesome due to the risks of insertional mutagenesis, host genetic alterations and germline transfers [7]. A safer route with plasmid delivery of OSKM was trialled by Hong et al., although a thorough analysis of the protocol is yet to be established [12]. An alternative route to the dedifferentiated, progenitor-like state that bypasses viral genetic handling was achieved with the cytokine-like protein dickkopf-3 (DKK3) [13]. Again, 4-day overexpression of adenoviral-delivered DKK3 followed by culturing in EGM-2 media supplemented with VEGF for 6 days, robustly reprogrammed human fibroblasts into functional iECs. Pluripotency marker expression (OSKM) did not change throughout the protocol. Showcasing their novel ex vivo circulation bioreactor system, the authors implanted their DKK3-reprogrammed iECs onto a decellularised aortic mouse graft, and after 5 days of culturing, observed an iEC monolayer surrounded by a multi-layered SMC wall. However, further in-depth analysis of cellular changes, marker expression patterns and EC-mural associations was not conducted.

The authors add to the growing evidence that the MET occurs when the cells move towards a progenitor-like state with VE-cadherin interactions acting as a prerequisite for further endothelial differentiation. Furthermore, the anti-angiogenic activity of microRNA (miR)-125a-5p through regulation of Stat3 (signal transducer and activator of transcription 3) was observed. Stat3 has been identified to improve the production of NO and prostacyclin (a vasodilator) and promote angiogenesis through interactions with VEGF [26]. Mimics of miR-125a-5p resulted in reduced activity of Stat3, whereas Stat3 silencing was associated with reductions in EC-specific gene expression and in vitro vascular formation. Interestingly, in the absence of Stat3, miR-125a-5p had no negative regulatory effects on EC differentiation [13].

While no in vivo studies were conducted, and the reprogramming efficacy is not yet established, this single-factor protocol by Chen and colleagues does set the foundations for safer and faster reprogramming approaches without extensive genetic manipulation. Moreover, adenoviral delivery harbours a reduced risk of host DNA integration and is unlikely to promote immune reactions to lentiviral counterparts [27]. Moreover, DDK3 is atheroprotective in ECs, which highlights a synergistic therapeutic benefit in reprogramming strategies [28].

### 2.2. Lineage-Specific Transcription Factors—The Advent of ETV2

ETV2 (E-twenty-six variant transcription factor 2) is well-established as a potent regulator of embryonic vascular development [29]. The factor is transiently expressed during embryogenesis, becoming virtually undetectable in postnatal ECs but significantly elevated following endothelial injury [30]. Indeed, both ETV2 knockout and prolonged exposure are associated with abnormal embryological vascular development [31,32]. Based on the above findings, Ginsberg et al. theorised that ETV2 alongside two other ETS-domain transcription family factors, FLI1 and ERG1, together with TGFβ inhibition, could induce expression of endothelial-specific genes and hence reprogramming of human mid-gestation lineage-committed c-Kit negative amniotic cells to iECs [14]. Indeed, transient lentiviral ETV2 expression induced an immature endothelial progenitor-like state which matured and ‘locked-in’ the vascular identity in response to continued FLI1 and ERG1 expression. Interestingly, they observed that stoichiometric expression of ETV2 with FLI1 and ERG1 was required to generate iECs from amniotic cells efficiently. Amniotic cell-derived iECs were functional and expandable with a similar transcriptome to human umbilical vein endothelial cells (HUVECs). Further constitutive signalling with the protein kinase AKT1 and transcription factor SOX17 has been shown to improve the activation of the global endothelial gene program [33]. However, Ginsberg and colleagues’ protocol failed to reprogram human postnatal cells, and interestingly, when repeated in another study, the addition of FLI1 and ERG1 observed a lower reprogramming efficacy in murine fibroblasts [34]. Moreover, the non-autologous implications of amniocentesis-derived therapy and the amniotic cell source have spurred a debate on the potential ethical and clinical limitations.

To circumvent the drawback of amniotic cells, Wong and Cooke used three other endothelial-specific transcription factors, FLI1, GATA2 and KLF4, alongside ETV2 to reprogram human neonatal fibroblasts into ECs, achieving a conversion efficacy of 16% for CD31-positive cells which was four times higher than the 5-factor protocol of ETV2, FOXO1, KLF2, TAL1 and LMO2 by Han and colleagues. ETV2 was identified as the most potent factor for reprogramming induction as systematically removing each transcription factor except ETV2 in 3-factor combinations only resulted in minimal reductions of CD31-positive cells. Neonatal fibroblasts were reportedly easier to reprogram than adult fibroblasts due to a more fluid epigenetic state. However, a study used modified mRNA encoding ETV2, FLI1, GATA2, and KLF4 to compare differences in angiogenic behaviour between reprogrammed cells derived from neonatal fibroblasts and dermal fibroblasts in patients with peripheral artery disease [35]. While successfully demonstrating reprogramming with modified mRNA, the study showed that neonatal and patient-derived iECs exhibited the same angiogenic behaviour, although this response was inconsistent among iECs derived from different patients. Further investigations are required to delineate differences between iECs generated from neonatal and adult cells.

Three studies reported ETV2 as an independent inductor of human adult fibroblast reprogramming, each with key insights into the endothelial reprogramming mechanism [16,17,18]. Morita and colleagues used a doxycycline-inducible system for lentiviral delivery of ETV2 to human adult fibroblasts, which after 15 days, were sorted for CD31-expressing iECs (3.5%). After suspension in Matrigel plugs and implantation into non-obese diabetic, severe combined immune deficiency mice, their iECs formed patent vasculature, which enabled erythrocyte circulation and expressed eNOS while almost 50% of iEC vessels associated with adjacent mural cells [17]. The authors further identified high levels of endogenous FOXC2 in human fibroblasts, interacting with ectopic ETV2 through a DNA binding site known as the FOX:ETS motif. Binding to the motif recruited factors which facilitated gene expression through post-transcriptional ETV2 alteration and modification of DNA methylation states of key endothelial genes. Different from the findings reported by Ginsberg et al. [14], ETV2 sufficiently induced endogenous FLI1 and ERG expression, and ectopic expression of the latter two factors was not required for generating iECs. An optimal differentiation efficacy was achieved with an intermediate ETV2 overexpression level, whereby deviations from this level negatively affected the reprogramming efficacy [17].

However, Morita and colleagues’ protocol used 15-day exposure to ETV2 with an extensive 50-day culturing period which researchers suggest may have forced the upregulation of endothelial markers without promoting cellular maturity [7]. With a similar methodology, Lee et al. failed to produce ECs from human fibroblasts and instead showed that transient ETV2 expression, independent of FOXC2, is obligatory for transdifferentiation and is reminiscent of embryogenic ETV2 expression patterns [18]. Initial lentiviral ETV2 induction generated early iECs characterised by mixed endothelial and fibroblast gene markers and low vWF and PECAM1 expression. The authors acknowledged that the residual fibroblastic features might be advantageous through synergistic paracrine modulation that matures and maintains early iECs. In a murine model, these early iECs were demonstrated to increase EC proliferation, neovascularisation and reduce limb ischaemia. However, whether the therapeutic benefits were attained through a paracrine effect or direct incorporation into existing vasculature is unclear. Early-to-late transformation of ECs was demonstrated through a 20-day transgene-free period followed by 6-day ETV2 exposure. Late-iECs observed increased silencing of fibroblastic signatures, low ETV2 and increased PECAM1 and NO production, suggesting more significant phenotypic similarities to primary mature ECs. Adding valproic acid (VPA) alongside the early-to-late iEC conversion improved the differentiation efficacy from 2% to 60%. In vivo studies demonstrated the incorporation of late-iECs into host vessels [18].

Morita et al. further observed an arterial endothelial specification with enhanced eNOS expression and mural-endothelial associations when their reprogrammed cells were constituted in Matrigel plugs containing mural cells and exposed to environmental changes such as shear stress [17]. However, without mural cell association, cells expressed venous markers suggesting a venous phenotype. In keeping with previous studies, these findings attribute the pivotal role of local microenvironments and dynamic mechanical forces in maturing and directing ECs to a distinct sub-phenotype. Further arterial specification was achieved with forskolin, a labdane diterpenoid extracted from the Coleus barbatus plant, which has shown several cardioprotective effects in early clinical studies [36,37]. Kim and colleagues first demonstrated that ETV2 activates cyclic AMP (cAMP) and exchange proteins directly activated by cAMP (EPAC) [16]. Subsequently, GTP-bound RAP1 protein is activated, which stimulates EC gene expression. Forskolin acted as a cAMP activator which promoted cAMP/EPAC/RAP1 signalling and observed a 3.2-fold increase in the number of CD31^+^/VE-cadherin^+^ ECs generated while promoting arterial-specific endothelial markers through Notch signalling. Forskolin treatment demonstrated further suppression of mesenchymal markers and improved neo-vascularisation in vivo compared to vehicle-treated cells. RAP1 activation by ETV2 was shown in another study to promote the development of durable lumens [38].

Solidifying the role of ETV2 in endothelial reprogramming, ETV2 was recently identified as a pioneer factor [29]. From studies on mouse embryonic fibroblasts (MEFs), the authors showed that ETV2 binds to nucleosomes and recruits BRG1, which then forms a complex acting to relax chromatin configurations of endothelial genes while recruiting other cofactors and increasing H3K27ac deposition. BRG1 maintained open chromatin formations allowing ETV2 to recruit other factors to activate endothelial gene expression. The authors also highlight the contribution of immune signalling in facilitating or repressing reprogramming based on their observation that a MEF cluster with greater inflammatory activity experienced greater upregulation of genes compared to a cluster with less inflammatory activity. Finally, ETV2 suppressed non-endothelial lineages by downregulating mesodermal genes [29].

### 2.3. Innate-Immune Activation—The Big Potential of Small Molecules

A research group has contributed significantly towards a paradigm shift away from pluripotency and lineage-specific transcription factors-based cellular reprogramming [9,39,40]. The researchers showed that innate immune activation is a facilitator and independent inductor of direct reprogramming. Innate immune activation via small molecules could induce a state of epigenetic plasticity, which is then amenable to differentiation with specific signals—a phenomenon termed ‘transflammation’ [39]. A key benefit of this approach is that viral vectors are avoided, and genetic manipulation is minimised. Through three progress studies, the research group reported three key findings: (1) innate immunity could be activated by targeting TLR3 (Toll-like receptor 3), which induces phenotypic plasticity through the (2) activation of the inducible-NOS (iNOS) signalling pathway and (3) the metabolic conversion from oxidative phosphorylation to glycolysis. Inhibition of iNOS and glycolytic switching reduced or inhibited endothelial differentiation. Through these processes, innate immune activation effectively increased DNA accessibility for reprogramming.

In their introductory study, an agonist of TLR3, polyinosinic:polycytidylic acid (polyI:C; PIC), induced a state of epigenetic plasticity, namely a global changes in epigenetic modifiers that increase the probability for an open chromatin state, in human fibroblasts which trans-differentiated into ECs in response to culture medium containing VEGF, BMP4, FGF, 8-Br-cAMP, and a TGFβ-inhibitor [9]. The resulting iECs shared similar functional and genetic characteristics to native ECs, with a conversion efficacy of 2% for CD31-expressing cells. Even though the iECs failed to incorporate with the endogenous vasculature in a peripheral artery disease murine model, a potential paracrine effect of iECs was observed to promote angiogenesis and improve tissue perfusion. The investigators further alluded to unpublished data suggesting the viability of targeting alternative receptors such as TLR4 and RIG-I (retinoic acid-inducible gene I) for innate immune activation.

The following study aimed to elucidate the cellular pathways that underline the above reprogramming process [40]. Upon agonist activity of TLR3, an extensive signalling cascade began with activation of iNOS mediated by NFkB. The end feature of the cascade through NO production or direct binding to iNOS was the destabilisation of PRC1, which maintains suppressive epigenetic marks such as the trimethylation of histone 3 at lysine 27 (H3K27me3) of the CD31 promoter region. Decreased H3K27me3 results in greater DNA accessibility for key transdifferentiation factors.

In their latest study on the topic, innate immune activation was coupled to metabolic activity with a switch from oxidative phosphorylation to glycolysis [39]. Specifically, iECs were generated by treating fibroblasts with induction medium containing Poly I: C (30 ng/mL) and EC georth medium supplemented with VEGF (50 ng/mL), bFGF(20 ng/mL), BMP4 (20 ng/mL), and 8-Br-cAMP (100 μM) for the 1st and 2nd week, respectively. The CD31^+^ iECs were sorted with flow cytometer and expanded in EC growth medium with SB431542 (10 μM). Mechanistically, by conducting multiple biochemical experimentation including Seahorse assay, the authors found that the metabolic changes occurred through numerous steps beginning with diverting some pyruvate and citrate away from the citric acid cycle and into the cytoplasm facilitated by the upregulation of mitochondrial citrate transporters (such as Slc25a1). Increased expression of nuclear ATP citrate lyase leads to the generation of nuclear acetyl-CoA. Acetyl-CoA then increases the activity of histone acetyltransferases which support histone acetylation and configures access to genes that promote transdifferentiation. The absence of alterations in fatty acid synthesis, upregulation of glycolytic enzymes by PIC and the induction of iNOS instead of eNOS provides compelling evidence that innate immune signalling independently contributed to the metabolic shift. However, the authors did observe metabolic heterogeneity in their population, which they suggested may be attributed to the metabolic heterogeneity of the starting fibroblast population. Ultimately, the iNOS pathway and glycolytic shift indirectly increase endothelial-specific gene expression by promoting an open chromatin composition. Further investigations into the metabolic shifts and mitochondria-nuclear signalling may identify more effective transdifferentiation strategies.

### 2.4. MicroRNA-Based Reprogramming

MicroRNAs (miRNA/miR) are increasingly observed to hold important roles in cell fate specification [41]. MiRNA was recently used to reprogram SMCs into iECs [19]. By identifying miRNA enrichment levels between ECs and SMCs, McCoy and colleagues demonstrated that transfection of EC-enriched miR-146a-5p and 181b-5p mimics, with the converse inhibition of SMC-enriched miR-143-4p and miR-145-5p could reprogram human coronary artery SMCs (CASMCs) into iECs. Their transdifferentiation protocol took place over 20 days with initial transfection followed by culturing in media consisting of EGM-2 (with a TFGβ inhibitor and 8-Br-cAMP), sorting for ICAM-1 positive cells and a final expansion phase. The iECs shared similar transcriptional and phenotypical profiles to HUVECs but not to CASMCs. In a murine hindlimb ischaemia model, mice transplanted with iECs experienced significantly faster perfusion times (142% faster) on day-11 post-injection and fewer toe loss (a sign of prolonged ischaemia) than HUVEC transplant recipients. The authors observed increased eNOS expression, which they believe reflects their angiogenic benefits. Furthermore, the upregulation of NOTCH1, JAG1 and DLL4 provides further evidence for the involvement of Notch signalling in SMC-to-EC conversion [19].

The 4-miR cocktail protocol confirms the viability of non-viral reprogramming approaches to generate iECs that are theoretically comparable to iECs generated through pluripotency and lineage-specific factors. However, there are a few areas for further improvement. Firstly, there may be better markers for iEC sorting than ICAM-1. Mature, quiescent ECs observe very low basal expression of ICAM-1, and its role is more robustly related to endothelial activation in pathogenic pathways, including atherosclerosis [42]. ICAM-1 is also expressed in other cells, such as macrophages and lymphocytes. Vascular cell adhesion molecule 1 is a more specific inflammatory marker for ECs, while CD31 and CD144 are generally considered highly reflective of mature ECs [43]. Secondly, the reprogramming efficacy is not clear which hinders any comparisons with other reprogramming approaches. Furthermore, several other miRs that could independently or synergistically induce reprogramming. For example, miR-539 and miR-582 are associated with SMC-to-EC communication to guide SMC development around ECs, while miR-125/126 have regulatory effects on TGFβ activity [19,44]. A cocktail of curated suppressive and expressive miR elements may yield high reprogramming efficacies. Further analysis of the effect of miR on disease states like atherosclerosis and diabetes will be essential to reduce the risk of adverse effects.

## 3. Smooth Muscle Generation

SMCs are found in the tunica media and help the artery withstand and regulate blood pressure by constricting and relaxing. Like ECs, several studies have reported using pluripotency and lineage-specific factors to reprogramme various somatic cells into induced-vascular smooth muscle cells (iSMCs) (Table 2; Figure 2). Human embryonic lung fibroblasts (HELF) were successfully reprogrammed into iSMCs with 4-day exposure to OSKM, followed by 4-day culturing in SMC differentiation medium [45]. The resulting iSMCs closely resembled native SMCs with the upregulation of SMC-specific markers such as SM22 (smooth muscle protein 22), calponin, SMA (smooth muscle actin-alpha), MYH11 (myosin heavy chain), SRF (serum response factor), MYOCD, and smoothelin. MYH11 and smoothelin are regarded as the best indicators of a mature SMC [46]. Other lineage-specific markers were not upregulated in the process, an important finding as some SM-specific markers are transiently expressed in other cell types like myofibroblasts. However, expression patterns of SM-specific miRs (e.g., miR-143) seen in mature SMCs and used as reprogramming factors by Mccoy et al. [19] were not investigated in this study. Nonetheless, fibroblast gene downregulation and contractile responses to KCl administration did suggest a mature and functional SMC phenotype of iSMCs. Although only 38% of cells were SM22-positive in this approach, antibiotic selection of cells with neomycin produced a pure population of SM22-positive iSMCs, as reported in this study [45].

Reprogramming HELFs to iSMCs identified a novel role of DKK3. During transdifferentiation, DKK3 binds to Kremen1 and activates canonical Wnt signalling, which then causes the translocation of β-catenin into the nucleus. Β-catenin then enhances the transcriptional activity of SM22 [45]. Interestingly, the reprogramming protocol presented in this study appears to be specific to HELFs and OSKM, as repeating the approach on skin fibroblasts observed no significant changes in SMC marker expression [45]. From their identification of DKK3′s involvement in reprogramming, Karamariti and colleagues further demonstrated adenoviral nucleofection of DKK3 as an independent inductor of reprogramming and readily transdifferentiated HELFs into iSMCs [47]. The iSMCs observed similar transcriptional and behaviour profiles to native SMCs. Analysis of the molecular pathway highlighted a different signalling cascade to OSKM-based iSMC reprogramming. DKK3 induced activity of the transcription factor ATF6 (activating transcription factor 6) which then increases transcription of TGFβ1. TGFβ1 expression increased SMC gene expression in fibroblasts. Furthermore, through several in-vitro and animal studies, the authors showed that DKK3-based reprogramming of both vascular progenitors and fibroblasts into iSMCs promotes stabilisation of atherosclerotic plaques (by suppressing inflammation and increasing SMCs)—highlighting a synergistic benefit of the protocol [47]. However, as DKK3 and TGFβ are both involved in reprogramming SMCs and ECs, the next challenge is identifying what pathways and molecules regulate each cell’s fate. Moreover, further work must elucidate whether different mediums or combinations of factors could better transform skin fibroblasts into SMCs and avoid the ethical implications of obtaining HELFs and the invasive nature of lung biopsies.

The conversion of human dermal fibroblasts (HDFs), a more ethically acceptable cell source, into iSMCs was achieved through the retroviral introduction of three factors, MYOCD, GATA6 and MEF2C (collectively known as MG2) with a conversion efficacy of 80% for MYH11-positive cells [48]. MYOCD is well-known as the master regulator of smooth muscle differentiation. However, the three-factor combination only induced endogenous MYOCD in mouse embryonic fibroblasts (MEFs), which the authors speculate may reflect an MYOCD-independent reprogramming pathway in human aorta media. Active suppression of Kruppel-like transcription factors (KLFs) is an alternative explanation for the finding. Identifying the regulators of MYOCD is a topic of ongoing research. However, evidence does demonstrate the KLF family’s repressive nature on MYOCD expression and their participation in the movements towards the synthetic-SMC phenotype [49,50]. Hence, high KLF5 expression in iSMCs compared to the control suggests partially reprogrammed cells. Further research needs to elucidate the role of MYOCD and its regulators in reprogramming. Suppressing KLF may facilitate the drive towards and maintenance of mature iSMCs.

## 4. Vascular Progenitor Cells

Several studies have reported small and rare colonies of progenitor cells found in different locations of the arterial wall, particularly the tunica adventitia [51]. These progenitors include endothelial, smooth muscle, haematopoietic, and multipotent stem/progenitor cells. For successful induction of a vascular progenitor, the final cell should exhibit high proliferative capacity, interact with the cellular matrix and readily differentiate into their respective cells. Similar to iECs and iSMCs, multiple cellular reprogramming methods have been reported to generate directly induced vascular progenitor cells (iVPCs) (Table 3; Figure 3).

### 4.1. Pluripotency Factor-Based Reprogramming

Forced expression of OSKM has directly reprogrammed adult fibroblasts into two types of bipotent VPCs [8,52]. The first approach by Kurian and colleagues used 8-day expression of OSKM and 8-day incubation in a mesodermal-induction-medium to generate CD34-positive progenitor cells capable of forming functional iECs and iSMCs [8]. CD34 is a characteristic marker of haematopoietic progenitor cells [56]. The benefit of progenitor cells in regenerative therapy was highlighted when an initial conversion efficacy of 20–60% for CD34-positive cells was increased to 400–1200% through expansion methods. Genetic characterisation of terminally differentiated cells indicated heterogeneous expression of markers, with arterial, venous, and lymphatic gene expression in iEC populations, and the presence of pericyte markers in iSMC populations. The authors attribute these findings to experimental alterations, poor downregulation of fibroblast signatures or variations in epigenetic plasticity between cells. Finally, the ability to repeat the protocol with episomal delivery of OSKM showcased the viability of non-integrating delivery methods that circumvent viral integration’s safety issues [8].

The second approach by Zhang et al. introduced OSKM into human adult dermal fibroblasts for 7-days to produce CD34-positive cells capable of differentiation into either iECs or induced-erythroblasts (iEBs) through culturing in endothelial and erythroblast differentiation mediums for 10-days and 4-days, respectively [52]. iEBs were identified by the erythroid marker CD235a. When assessed in vivo, new vessels from human CD34^+^ progenitor cells readily communicated with the existing murine vessel and contained murine erythrocytes. The release of differentiation signals like VEGF in ischemic tissue and erythropoietin in circulating blood is a potential mechanism for differentiating transplanted CD34^+^ progenitors into iECs or iEBs. Furthermore, the transcription factor SOX17 (SRY-box transcription factor 17) was identified as a ‘tuneable rheostat-like switch’ whereby overexpression favoured the endothelial lineage compared to depletion, which promoted the erythroblast fate [52]. Thus, SOX17 is a regulator of endothelial development and a cell-fate decider. Another key finding from this study is the increased telomerase activity upon de-differentiation, which marks a crucial therapeutic benefit where cells could undergo prolonged proliferation.

The generation of cardiac progenitor cells (iCPCs) with tripotent potential for ECs, SMCs and cardiomyocytes (CMs) was achieved through a three-stage protocol [53]. (1) Murine fibroblasts were first exposed to OSKM and JI1 (Jak inhibitor 1) for 5-days followed by (2) 2-day treatment with JI1 and an activator of canonical Wnt signalling (CHIR99021). The transition to a progenitor was completed with (3) 14-day culturing in a medium containing BMP-4, Activin A, CHIR99021 and SU5402 (an inhibitor of VEGF), collectively known as BACS. The expression of markers FLK-1 and PDGFR-α indicated the progenitor state. The terminal progenitor cells were restricted to the cardiovascular lineage, displayed significant self-renewable capacity, expanded more than 10^10^-fold and were genetically stable for more than 18 passages. Although aimed at cardiac regeneration, this protocol is advantageous for vascular repair. For example, progenitors may support coronary artery regeneration, and the CMs can improve cardiac function. In vivo transplantation of iCPCs into mice resulted in the generation of cells of the 3 lineages, with 90% of iCPCs undergoing differentiation to form SMCs, CMs and ECs in an approximated ratio of 6:3:1. In a myocardial infarction murine model, functional improvements such as reduced scar size were observed. Whether the cells directly promoted the therapeutic benefits or indirectly through paracrine modulation is unclear and requires further investigation. A caveat with the authors’ protocol was that not all iCPCs were tripotent, with 45.5%, 22.7%, and 31.8% of them being unipotent, bipotent or tripotent, respectively. It will be interesting to investigate whether variations to the reprogramming protocol could dictate which potency level and lineage(s) is derived). The mechanism underlying BACS in facilitating de-differentiation to a progenitor is another subject for future research.

### 4.2. ETV2

In the previous discussion of EC generation, ETV2 transduction reportedly led to the development of a progenitor-like state which was amenable to differentiation under specific cultures [14]. However, the progenitor cells needed to be better characterised, and only their lineage derivates were transplanted into mice for in vivo studies. Recently, in the absence of differentiation media, ETV2 successfully generated induced-endothelial progenitor cells (iEPCs) [54,57]. Through either lentiviral delivery or using modified mRNA, ETV2 overexpression for 14-days propagated expression of CD31-positive iEPCs, which robustly formed capillary-like networks within Matrigel plugs and improved tissue perfusion in a murine hindlimb ischaemia model. Hypoxic conditions (5% oxygen) improved the lentiviral-based reprogramming efficacy 6-fold from 1.21% under normoxia to 7.5% for CD31-positive cells with hypoxia. Notably, a much lower efficacy of 3.1% under hypoxic conditions was observed with modified ETV2 mRNA delivery. Nevertheless, following sorting, both protocols achieved almost pure CD31-positive cell populations.

A significant finding of the ETV2-based approach was using an alternative medium for culturing progenitor cells [54]. Thus far, most studies discussed have used 10% fetal bovine serum (FBS) to supplement the differentiation media. The use of FBS has several ethical implications, as some see the procedure to derive the serum from unborn foetuses as inhumane. Moreover, animal proteins may lead to adverse immune reactions and animal-to-human viral transmission when transplanted [58]. Hence, the researchers showed that platelet-rich plasma derived from a patient’s peripheral blood (PRP) could be an alternative to FBS in various mediums [54]. Non-viral factor delivery and animal-free culture present a breakthrough in the safety of reprogramming. However, the investigators did not attempt a thorough genetic characterisation of cells derived from iEPCs. Moreover, the proliferation capacity of iEPCs was not assessed but did indicate that fibroblasts proliferated prior to ETV2 administration.

In another study, ETV2 and FLI1 co-expression induced bipotent VPCs (iVPCs) identified by the expression of the CD144 marker [55]. The CD144-positive iVPCs stably expanded for 25 passages with an average doubling time of 40 h. The iVPCs differentiated into functional iECs and iSMCs when cultured in respective differentiation media. In a murine hindlimb ischaemia model, transplantation of the iVPCs resulted in improved tissue perfusion. Further research should explore the molecular pathways in which ETV2 and FLI1 facilitate reprogramming to iVPCs. It is worth noting that in the above two papers, CD31 and CD144 were used to select iVPCs. However, these markers are also present on mature ECs. Hence, one may question the validity of the iVPCs phenotype. Using more specific markers or a full panel of genetic markers to confirm the cellular phenotype will ensure improved reliability and complete characterisation of generated cells.

## 5. Tissue Engineered Vascular Grafts

TEVGs are biodegradable, cell-seeded scaffolds which can be implanted into a host to develop over time into functional vessel-like structures [59]. The central idea with TEVGs is that the scaffold enables cell anchorage and maturation, stimulates extracellular matrix (ECM) deposition and host cell tissue repair, which can maintain the robust vascular structure after the polymer scaffold biodegrades. The value of TEVGs in direct reprogramming research is two-fold. Firstly, TEVGs can be used to investigate reprogrammed cell function within the microvascular environment. For example, induced cells could be evaluated for their response to various factors, including the mechanical strain from high-pressure pulsatile blood flow (through bioreactor systems), circulating hormones, cytokines, oxygenation, immune activity and ECM deposition. Moreover, CVD’s underlying disease processes and risk factors (e.g., high blood glucose and lipid levels) can alter vascular cell function. Hence, the response of reprogrammed cells to both normal and diseased physiology is paramount for safety assessments before clinical translation. Secondly, contemporary autologous and artificial grafts fail to achieve long-term survivability and sustainability in replacing diseased vessels with small diameter (<6 mm) [60,61]. By combining direct reprogramming with innovative bioengineering materials and manufacturing methods, patient-specific and genetically augmented vascular cells could be grafted onto scaffolds with low immunogenic properties. iPSCs-seeded TEVGs have already demonstrated good integration and therapeutic benefits in animal studies, and one may sufficiently assume that the benefits of direct reprogramming compared to iPSCs may have greater advantages with TEVGs [62,63]. Moreover, while in vivo methods such as directly injecting reprogrammed cells can be therapeutically beneficial in medium to long-term treatment plans, short-term or urgent clinical requirements may be less feasible. Hence, ‘off-the-shelf’ or rapidly developed TEVGs will provide holistic coverage of any therapeutic requirements. TEVG is a dynamic interdisciplinary field with an overwhelming number of strategies, from improving the fundamental mechanical and chemical properties to the detailed microscopic control of surface morphology and augmentation of post-transplant thrombogenic processes [4,64,65,66]. Since TEVG is a complex topic outside the scope of the present view, the following section provides the reader with a brief exploration and exciting examples of where direct reprogramming can be applied to TEVGs. Herein we explore using decellularised tissue, 3D bioprinting and scaffold-based systems to generate TEVGs for research and therapeutic purposes (Figure 4).

### 5.1. Decellularised Tissue

Decellularised tissue grafts typically arise from animal or cadaveric vessels that undergo cell and nuclear removal to leave behind the natural ECM scaffold [65]. Decellularised grafts have several advantages: reduced host-immune responses; presence of natural biochemical and biomechanical properties; presence of bioactive substances that facilitate the migration of endogenous cells and progenitors; observe high biodegradability without the release of toxic products, and can be re-populated with patient-derived cells [67]. Disadvantages include incomplete decellularization, loss of some properties through the decellularization process, difficulty achieving complete recellularisation and mismatch between graft degradation rates and the tissue regeneration rate [67,68]. Several papers have elegantly documented the advantages and disadvantages of decellularised grafts [69,70,71].

Several authors have provisionally examined their directly reprogrammed cells on decellularised aortic grafts (DAG). Margariti et al. seeded their PiPSC-Ecs onto the DAG, which was followed by culturing in a bioreactor system that emulated physiological blood flow. The cells aggregated into the typical vessel morphology with Ecs in an elongated and orientated pattern with patent lumens [10]. Double-seeded PiPSCs demonstrated the ability to generate an EC monolayer and multiple SMC layers, which improved vessel stability. Hong et al. injected SMC-derived iECs on the luminal surface of the DAG with primary SMCs on the outside and observed similar results as Margariti and colleagues but further exemplified the potential for the same cell source to develop both the endothelial and smooth muscle components [12]. Karamariti et al. went further and engrafted their double-seeded PiPS-derived Ecs and SMCs into mice [45]. Their graft observed patent vasculature and a survival rate of 60% 21-days post-transplantation compared to unseeded DAG and fibroblast-seeded DAG, which experienced rupture and luminal occlusion, respectively. In an alternative to DAG, Kim et al. observed complete endothelialisation and maintenance of endothelial features on a decellularised rat liver scaffold with their iECs [16]. While the studies mentioned confirm the viability of reprogrammed cells in engineered grafts, they fail to analyse key vascular physiology markers such as contractile function and ECM formation. For example, Ji et al. observed dilatory and constriction responses to vasoactive stimuli (flow rate changes, phenylephrine and acetylcholine) in their TEVG composed of iSMCs transdifferentiated from endothelial progenitors [72]. Such methods are more conducive to evaluating cell and tissue function in 3D microenvironments than traditional 2D dish-based analysis.

One limitation of decellularised grafts is the difficulty replanting cells due to the complex ECM architecture [68]. A solution is to use solubilised decellularised grafts to create 2D coatings and 3D hydrogels to improve cell adhesion and maturation [73]. 3D hydrogels may provide greater recapitulation of the mature ECM microenvironment in vivo [74]. Jin and colleagues demonstrated that brain ECM-based 2D coatings and 3D hydrogels not only enhanced conversion efficacies of induced-neuronal cells transdifferentiated from fibroblasts (through plasmid-delivery of reprogramming factors), but further propagated cell maturation, gene expression and attained significant therapeutic benefits in animal studies [73]. The authors further observed only marginal differences between their human brain-ECM batches, which were nevertheless all conducive to successful reprogramming, highlighting that batch-to-batch variation studies for decellularized arterial grafts are imperative to understand future reproducibility. Another key takeaway is that their hydrogel can successfully overcome the low conversion efficacies experienced by non-viral delivery systems [73].

### 5.2. 3D Bioprinting

3D bioprinting is an emerging approach to directly print cells and supporting material (referred to as bioink) to create 3D structures with high precision and curated spatial distribution [75]. Liguori et al. used a DAG hydrogel to bioprint the tunica media of a small-diameter blood vessel [76]. The resulting tissue was densely populated and viable over their 7-day observation period. Interestingly, the hydrogel provided sufficient cues to drive spontaneous differentiation of stem cells to SMCs. Ho and Hsu showcased an approach to combine direct reprogramming and 3D bioprinting. The authors first generated a thermosensitive and waterborne polyurethane gel which contained human fibroblasts and the neural direct reprogramming inductor, FoxD3. The authors then demonstrated that the extrusion pressure generated when extruding the fibroblast and FoxD3-laden polyurethane gel through a syringe needle of a 3D bioprinter was sufficient to transfect fibroblasts with FoxD3 and consequently generate neuron-like cells with a conversion efficacy of 15.6%. Thus, this study provides a non-viral and in-situ approach for reprograming cells and synergistically generating a 3D graft [77]. The addition of biomechanical stimuli post-bioprinting can further mature and functionalise cells, as seen with bioprinted iPSC-cardiomyocytes, which attained greater sarcomere length and contractile forces after mechanical stretching [78]. Similarly, drug-releasing microspheres could be added to the bioink or post-bioprinting to provide constitutive signalling (such as EC differentiation signals) that may mature and maintain the induced-cell phenotype [79].

### 5.3. Scaffold-Based Grafts

Scaffold-based grafts use synthetic and natural macromolecular structures to facilitate tissue regeneration [65]. Different properties such as fibre material, fibre and pore size, mechanical stiffness and degradation rates can be tightly controlled to develop niche microenvironments to regulate cell fate and function. Sato et al. demonstrated the development of a 3D synthetic scaffold that facilitated direct reprogramming of fibroblasts into induced osteoblasts (iOBs) [80]. After transducing fibroblasts with their reprogramming inductors, the cells were placed and cultured on a fibronectin coated nanogel-cross-linked porous-freeze-dried (NanoCliP-FD). iOBs readily adhered to the scaffold due to the large pores and deposited large amounts of bone matrix with significant bone regeneration observed in vivo. The authors further state that dehydration of the gel into a matrix would improve storage and transport, with the return to a gel undertaken prior to transplantation. Controlling pore size is important for vascular cells. Smaller pores result in greater SMC populations in the lumen with more ECM deposition. If the pore is too small, cells may not readily seed on the graft [81]. In another study examining iPSC-neural crest cell seeded scaffolds, it was observed that the 3D scaffold selectively removed undifferentiated iPSCs through cell confinement which also matured induced-cells to achieve an almost 38-fold increase in cell survival compared to dissociated cells [82]. This finding is exciting and highlights the role of scaffolds to organise isolated cells, improve conversion efficacies and to reduce the risk of tumorigenesis in direct reprogramming approaches. Similarly, mesenchymal stem cells readily differentiated to SMCs with stiffer pectin hydrogel scaffolds, while the EC lineage was promoted with softer scaffolds [79]. Other scaffold materials such as poly (lactide-co-glycolide)/polyethylene glycol [83] and polylactic acid [84] have also observed remarkable success with directly reprogrammed cells.

## 6. The Ethics of Direct Reprogramming

The fundamental ethical advantage of direct reprogramming is the reduced focus on human embryonic stem cells (hESCs). As somatic cells are the cell source, there is no need for human blastocysts to be used or produced, which many regards as unethical due to the risk to and loss of potential life [85]. Moreover, therapeutic cloning becomes redundant if direct reprogramming can achieve the same results [86]. However, there is still a residual ethical conflict as the use of hESCs cannot be disregarded entirely. It is worth noting that direct reprogramming research is still in its infancy. Further in-depth and side-by-side comparisons of hESCs and their derivatives will need to be conducted to identify and understand the role of various transcription factors and regulators. hESCs are a natural derivation of embryogenic physiology compared to what may be seen as ‘unnatural’ with iPSCs and reprogramming. Therefore, hESCs will better assess how genes, transcription factors, molecules, materials, markers and mediums are employed during natural de-differentiation and differentiation processes. Thus, in the short term, hESCs are required to develop standards to compare cells and understand programming pathways. However, once differentiation and reprogramming mechanisms are fully validated, hESCs may be wholly replaced. Similarly, reprogramming approaches using amniotic or human embryonic cells will experience challenges associated with gaining informed consent and the lengthy process of gaining approval and oversight from ethical and regulatory institutions. In the meantime, the most sensible and uncontroversial sources for cellular reprogramming are adult skin fibroblasts, peripheral blood cells, urine epithelial cells, and cells from bariatric or metabolic surgery.

## 7. Current Challenges & Future Perspectives

The review thus far has identified exciting outcomes and outlooks for direct reprogramming approaches. However, several notable challenges remain and need to be addressed with an outlook for clinical translation. In the following section, we highlight key challenges and provide potential avenues for overcoming them.

### 7.1. Factor Identification and Reprogramming Efficacies

The biggest challenge for direct reprogramming is identifying a single or group of factors that efficiently propagate reprogramming. Traditional factor identification strategies have used trial-and-error methods, which are tedious, time-consuming, and slow [34]. Thus, studies have primarily used well-known pluripotency factors or master regulators such as ETV2. However, even with the use of such potent regulators, reprogramming strategies have encountered hugely varied conversion efficacies ranging from as low as 1% to >90%, which are dependent on a complex network of factors such as the specific reprogramming factor(s) used, their exposure period, molecular boosters, microenvironmental aids (e.g., shear stress and hypoxic environments) and level of inflammatory activity. While direct reprogramming strategies are still in their infancy, it is still valuable to explore a range of factors outside the well-known regulators as there may be better inductors of reprogramming.

To rapidly explore other transcription factors and combinations, several groups have developed computational prediction systems which use gene expression and regulatory network data to identify factors that promote the desired lineage conversion [87]. Such systems enable rapid screening of factors and essentially fall into two categories: transcription factor identification, which aims to identify transcription factors involved in conversion, and transcription factor perturbation which is a simulation of the effects of transcription factors on a cell [88]. Examples of the former include TRANSDIRE (TRANS-omics-based approach for DIrect REeprogramming) by Eguchi et al., which predicts pioneering factors and transcription factors for a range of cell conversion, and Mogrify by Rackham et al., which produces an eight-list rank of top transcription factors that potentially regulate the conversion from specific initial cells to final targeted cells [89,90]. Ronquist et al. demonstrated a perturbation system that further identifies the amount and induction time of the required transcription factors for a specific conversion [91]. As technology advances, computational systems may incorporate non-transcription factors to identify other small molecules and miRs capable of successfully reprogramming cells. Likewise, it is vital to explore molecules, culture conditions and mediums that can boost the reprogramming efficacy, as seen with VPA and forskolin. For further information on computational prediction algorithms and future perspectives, the reader is directed to several in-depth reviews [87,88,92,93].

### 7.2. Heterogeneity of Derived Cell Populations

Very few studies examined the specific subtypes of progenitors and differentiated cells produced. Using the wrong subtype may have unforeseen consequences. For example, the endothelium in medium-large arteries is continuous, whereas in the liver and kidney, the endothelium may be fenestrated or sinusoidal owing to their respective functions. Developing iECs that form a continuous monolayer is crucial for arterial function. Similarly, SMCs display significant heterogeneity in normal anatomy and are characteristically different depending on their embryonic origin (e.g., neural crest, mesodermal) and local factors (biochemical, extracellular matrix and physical) [94,95]. Thus, they may experience differing production of growth factors, morphology and resistance to vascular disease [96]. For example, cell proliferation and DNA synthesis are increased by TGF-β1 in neural crest-SMCs but decreased in mesoderm-SMCs [97]. In addition, venous SMCs tend to be less differentiated, hold greater proliferative capabilities and have higher tendencies to develop atherosclerosis in venous grafts compared to the arterial phenotype [98]. Consequently, each artery possesses a distinct mosaic pattern of SMCs geared towards their specific regional function. Thus, research should understand whether transplanting venous or lymphatic phenotypes of iSMCs and iECs can achieve the desired therapeutic benefits in arterial structures. Identifying markers may help elucidate specific sub-phenotypes better and help identify molecules that direct cells to a specific phenotype.

Furthermore, quantifying arterial and venous markers and morphological changes to categorise heterogeneity is inadequate; protocols to screen for heterogeneous functional responses are needed. For example, using lentiviral integration of GCaMP6f, a sensitive Ca2^+^ sensor, into SMCs, Cuenca et al. could observe and quantify dynamic intracellular calcium changes and analyse vasoactive heterogeneity [99]. Similar technologies that can assess variations in acetylated-LDL uptake and NO excretion in an EC population will give more holistic interpretations of functional heterogeneity.

Cellular tracking technology may also be helpful for differentiating reprogrammed cells from the starting population, and tracking cellular changes and cell division during reprogramming. Several methods have already been reported [100,101]. ScarTrace is a single-cell sequencing approach which adds fluorescent tags to cells enabling tracking in multiple locations on an organism [102]. Similarly, the Cre-loxP recombinase system activates GFP (green fluorescent protein) expression in specific cells and permits tracking their off-spring in vivo, which may prove to be useful in tracking the differentiation of induced-vascular progenitor cells [100].

### 7.3. Factor Delivery Systems & Viral Integration

With the end goal of clinical translation, factor and molecule delivery systems should be safe and target the desired cell type and, where appropriate, the specific genes. Lentiviral delivery systems are commonplace in reprogramming strategies for their long-term gene expression and packaging capacity for large transgene sequences [27]. However, they possess several safety risks due to unpredictable transgene insertion, which can cause insertional mutagenesis and inadvertently activate proto-oncogenes or silence tumour suppressors, increasing cancer risk [103]. Other non-integrating viruses like adenoviruses and Sendai viruses are reportedly safer as they harbour low immunogenic properties and rarely insert into host DNA. However, adenoviruses possess smaller insert sizes and are difficult to direct to specific cells [27]. Thus, given the advantages and disadvantages of different viral vectors, research should continue exploring both lentiviral and non-lentiviral vectors and analyse what developments and modifications can improve their safety profiles, gene expression stability and packaging capacities. For example, monitoring technology for replication-competent viruses and protocols for reducing infectious virus particles can lower the risk of mutations [104].

CRISPR/Cas9 is a growing gene activation approach that has garnered considerable attention in reprogramming various somatic cells into iPSCs [105,106,107]. Recently, CRISPR was used to endogenously activate gene promoters of the GNT gene, which successfully reprogrammed fibroblasts into cardiovascular progenitors [108]. This gene editing approach is a safe alternative as it avoids integrating viral vectors and exogenous expression of pluripotency factors. Other transgene-free vectors, such as small molecules (DKK3, TLR3 agonists and miRs) and plasmids, are viable alternatives. However, their use is rare, and their efficiency and safety profiles are still under review [12].

### 7.4. Tumorigenicity Risk

No study observed the development of tumours after transplanting the induced cells into mice over the respective observation periods. While this suggests a low tumorigenicity risk, the risk is not absent. For example, one study did find that one OSKM-derived induced-vascular progenitor clone did form a tumour [20]. Hence, sorting and purification methods must be robust and reliable. Although differentiated cells may pose a lower tumour risk to progenitors, we must not forget that reprogrammed cells undergo gross metabolic, chromosomal, genetic and epigenetic alterations. Thus, even if differentiated cells are transplanted, there is an unknown risk that they harbour abnormalities which adversely affect cell function or have a preponderance towards cancer formation. For example, a dysfunctional cell could induce pathogenic changes, such as promoting thrombosis through overexpression of adhesion molecules. Studies should look for multi-year assessments of tumorigenicity in longer-living animals (e.g., pigs and sheep) and closer animal relatives to humans. Furthermore, there is an inherent tumour risk with using pluripotency factors, notably c-MYC (in OSKM), which has been shown as both dispensable and a facilitator of lineage-conversion [109,110]. As further research occurs, we may find efficient non-oncogenic factors for direct cellular reprogramming.

### 7.5. Recapitulating Disease

There is growing evidence that direct reprogrammed cells and iPSC-derivates may retain disease signatures [111,112]. This finding has both positive and negative consequences. On the one hand, reprogrammed cells could allow in vitro modelling of patient-specific diseases and drug responses. Strategies could be employed to identify how different reprogramming approaches may respond depending on the underlying disease process. On the other hand, we are presented with another significant challenge: how to erase disease signatures successfully. This research topic will become much more pronounced as direct reprogramming advances towards clinical application.

### 7.6. 2D In Vitro Analysis vs. 3D Microenvironments

The studies analysed thus far are based mainly on 2D analysis, which fails to recapitulate in vivo 3D microenvironments. Several direct reprogramming studies have highlighted improved conversion efficacies and cellular maturation with mimics of 3D microenvironments compared to 2D in vitro studies which may underestimate reprogramming efficiencies [82,113,114,115]. We have already discussed how TEVGs can act as both investigative and therapeutic platforms by emulating specific 3D microenvironments to facilitate reprogramming and cell viability. Another avenue is vascular-on-chip models (VoCs), which provide a dynamic platform to test physiological and pathogenic processes on reprogrammed cells. For example, one could assess the endothelial barrier function of iECs or the haemostatic, immune and inflammatory responses to endothelial injury, as already exemplified by several groups [116,117]. Cuenca et al. used a VoC model and observed how the co-culturing of iSMCs and iECs with each other, adding mural cells and mechanical forces propagated further cell maturation, network self-assembly and stabilisation [99]. VoCs also hold potential for experimental analysis on the effect of drugs on diseased arterial cells and reprogrammed cells in a highly individualised manner whereby cells and blood can be taken directly from a patient. Such approaches provide holistic interpretations of reprogrammed cell activity under in vivo settings and are the next logical step towards clinical application and personalised medicine.

## 8. Conclusions

Direct reprogramming has resulted in a paradigm shift away from traditional stem cell reprogramming approaches. By introducing pluripotency factors, lineage-specific transcription factors, and small molecules, alongside curated culture conditions and mediums, somatic cells have successfully converted into iECs, iSMCs, and iVPCs without a pluripotent intermediate step. Emerging evidence has collectively shown huge therapeutic potentials for these vascular cells directly reprogrammed from other somatic cells (mainly fibroblasts) (Figure 5). Data from the preclinical studies discussed in this review have confirmed the potential for significant therapeutic benefits from increased angiogenesis and reduced ischaemia by directly incorporating reprogrammed cells with endogenous vasculature and paracrine modulation. Importantly, the advent of computational processing for reprogramming factor identification, TEVGs for disease modelling and therapeutics, 3D platforms such as VoCs, and more sophisticated gene/factor delivery systems will provide a giant leap in direct reprogramming research. However, several challenges still need to be addressed, notably the low conversion efficacies, heterogeneity of reprogrammed populations and the risk of integrating gene delivery systems such as lentiviruses. Further work should aim to identify effective reprogramming factors and comprehensively elucidate their underlying molecular pathways. As research continues and the standardisation of materials, cultures, and protocols becomes widespread, translation to human studies and clinical application for CVD treatment may appear sooner rather than later.

## Figures and Tables

**Figure 1 jfb-14-00021-f001:**
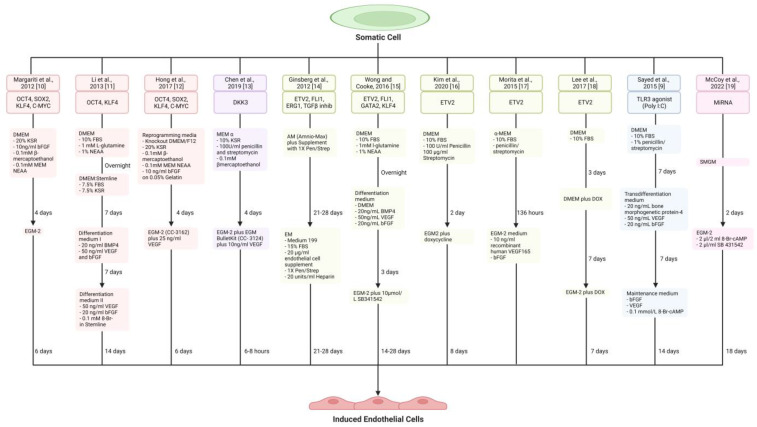
Schematic diagram showing key reprogramming factors, induction medium, and duration used for creating induced endothelial cells (iECs) in the different studies. Colour code: Red, Pluripotency factor-based reprogramming; Purple, DKK3-induced cellular reprogramming; Green: Lineage-specific transcription factor ETV2-based reprogramming; Blue, Innate-immune activation-induced cellular reprogramming; Pink, MicroRNA-based reprogramming. bFGF, basic Fibroblast Growth Factor; 8-Br-CAMP, 8-Bromo-cAMP; cAMP, cyclic AMP; BMP4, Bone Morphogenetic Protein 4; DKK3, Dickkopf WNT Signalling Pathway Inhibitor 3; DMEM, Dulbecco’s Modified Eagle Medium; EC, Endothelial Cell; EGM-2, Endothelial Growth Medium-2; EM, Endothelial Medium; ERG1, Early Growth Response Protein 1; ETV2, ETS Variant Transcription Factor 2; FBS, Fetal Bovine Serum; FLI1, Friend Leukaemia Integration 1 Transcription Factor; GATA2, GATA-binding factor 2; KSR, Knockout Serum Replacement; MEM, Minimum Essential Medium; miRNA, MicroRNA; NEAA, Non-Essential Amino Acid; OSKM, OCT4 (Octamer-Binding Transcription Factor 4), SOX2 (SRY-Box Transcription Factor 2), KLF4 (Kruppel-Like Factor 4) and C-MYC (c-Myc proto-oncogene protein); Pen, Penicillin; Poly I:C, Polyinosinic-polycytidylic acid; SB341542, TGF-β Receptor Kinase Inhibitor; SMGM, Smooth Muscle Cell Growth Media; Strep, Streptomycin; TGFβ, Transforming Growth Factor Beta; TLR3, Toll-Like Receptor 3; VEGF, Vascular Endothelial Growth Factor.

**Figure 2 jfb-14-00021-f002:**
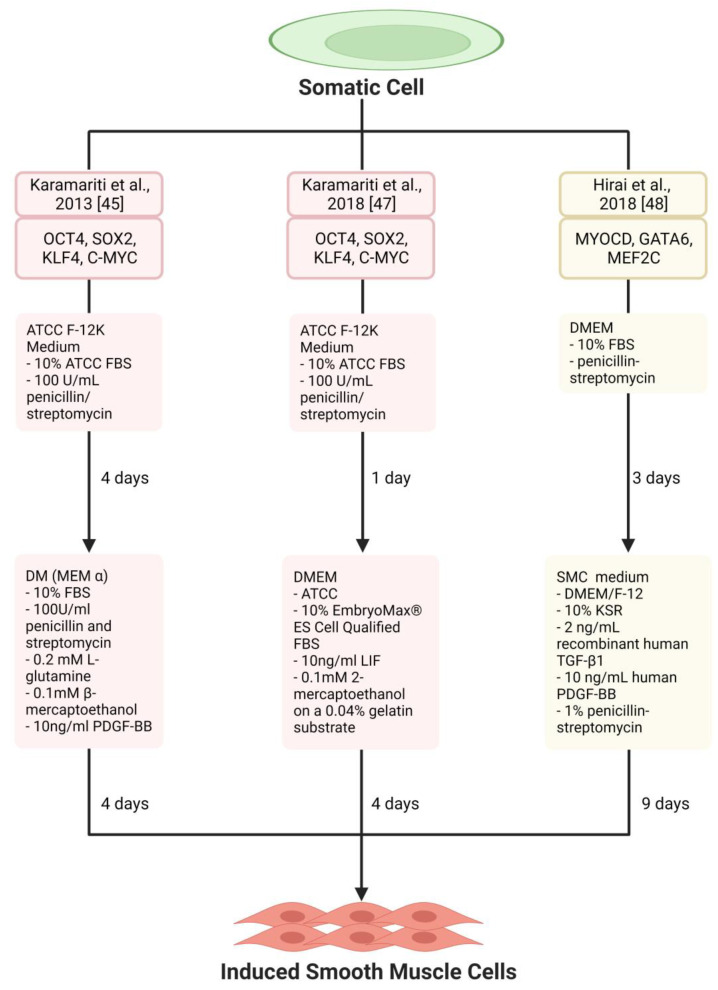
Schematic diagram showing key reprogramming factors, induction medium, and duration used for creating induced smooth muscle cells (iSMCs) in the different studies. Colour code: Red, Pluripotency factor-based reprogramming; Yellow, Lineage-specific transcription factors-based reprogramming. ATCC, ATCC medium; ATCC F-12K, ATCC-Kaighn’s Modification of Ham’s F-12 Medium; DKK3, Dickkopf WNT Signalling Pathway Inhibitor 3; DM, Differentiation Medium; DMEM, Dulbecco’s Modified Eagle Medium; GATA6, GATA-binding factor 6; KSR, Knockout Serum Replacement; MEF, Mouse Embryonic Fibroblasts; MEF2C, Myocyte-Specific Enhancer Factor 2C; MEM α, Minimum Essential Medium α; MYOCD, Myocardin; OSKM, OCT4 (Octamer-Binding Transcription Factor 4), SOX2 (SRY-Box Transcription Factor 2), KLF4 (Kruppel-Like Factor 4) and C-MYC (c-Myc proto-oncogene protein); PDGF-BB, Platelet-derived growth factor-BB; SMC, Smooth Muscle Cell; TGFβ1, Transforming Growth Factor Beta 1.

**Figure 3 jfb-14-00021-f003:**
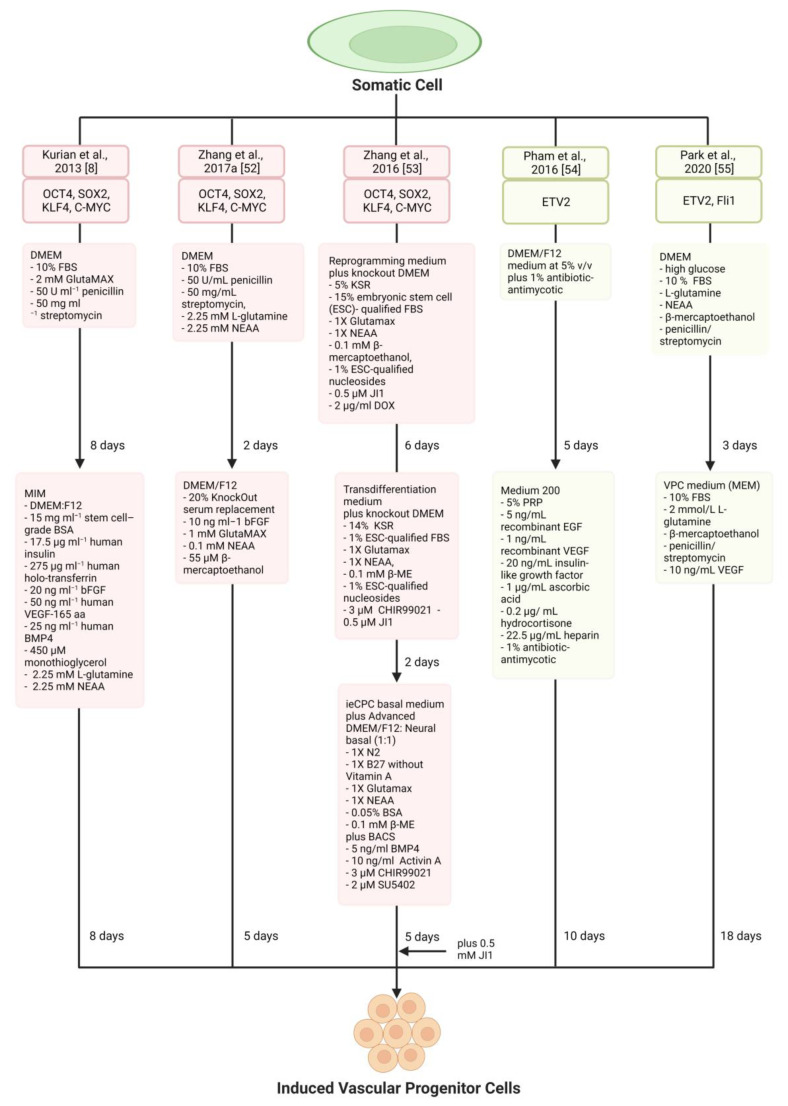
Schematic diagram showing key reprogramming factors, induction medium, and duration used for creating induced vascular progenitor cells (iVPCs) in the different studies. Colour code: Red, Pluripotency factor-based reprogramming; Green: Lineage-specific transcription factor ETV2-based cellular reprogramming. BACS, BMP4, Activin A, CHIR99021, SU5402; bFGF, basic Fibroblast Growth Factor; β-ME, β-mercaptoethanol; BMP4, Bone Morphogenetic Protein 4; BSA, Bovine Serum Albumin; CHIR99021, Glycogen Synthase Kinase 3 Inhibitor; DMEM, Dulbecco’s Modified Eagle Medium; DOX, Doxycycline; EGF, Endothelial Growth Factor; FBS, Fetal Bovine Serum; Fli1, Friend leukaemia integration 1; JI1, Jak Inhibitor 1; KSR, Knockout Serum Replacement; OSKM, OCT4 (Octamer-Binding Transcription Factor 4), SOX2 (SRY-Box Transcription Factor 2), KLF4 (Kruppel-Like Factor 4) and C-MYC (c-Myc proto-oncogene protein); MEM, Minimal Essential Medium; MIM, Mesodermal Induction Medium; NEAA, Non-Essential Amino Acid; VEGF, Vascular Endothelial Growth Factor.

**Figure 4 jfb-14-00021-f004:**
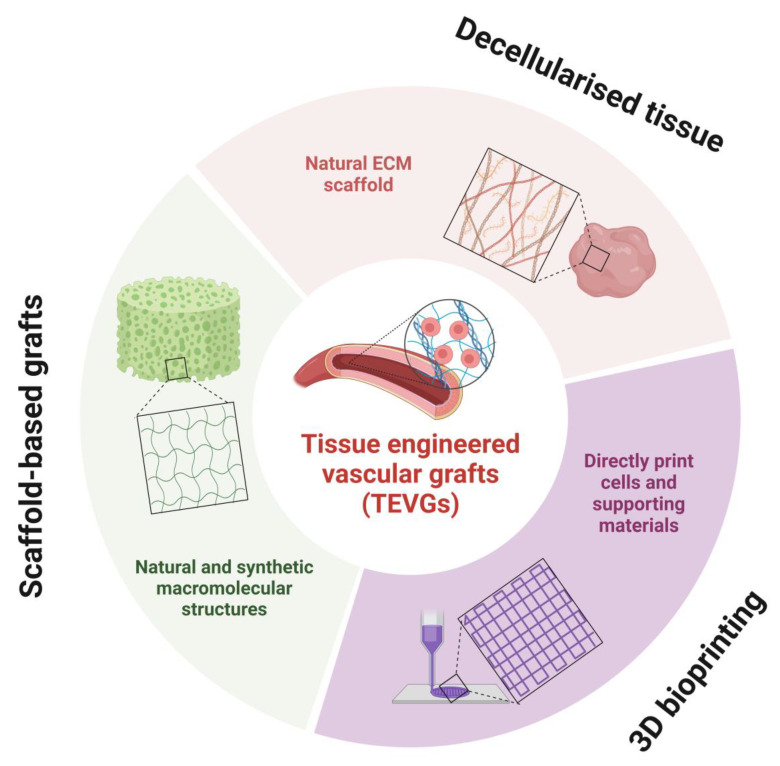
Tissue engineered vascular grafts (TEVGs) generated using decellularized tissue, 3D bioprinting and scaffold-based grafts, respectively.

**Figure 5 jfb-14-00021-f005:**
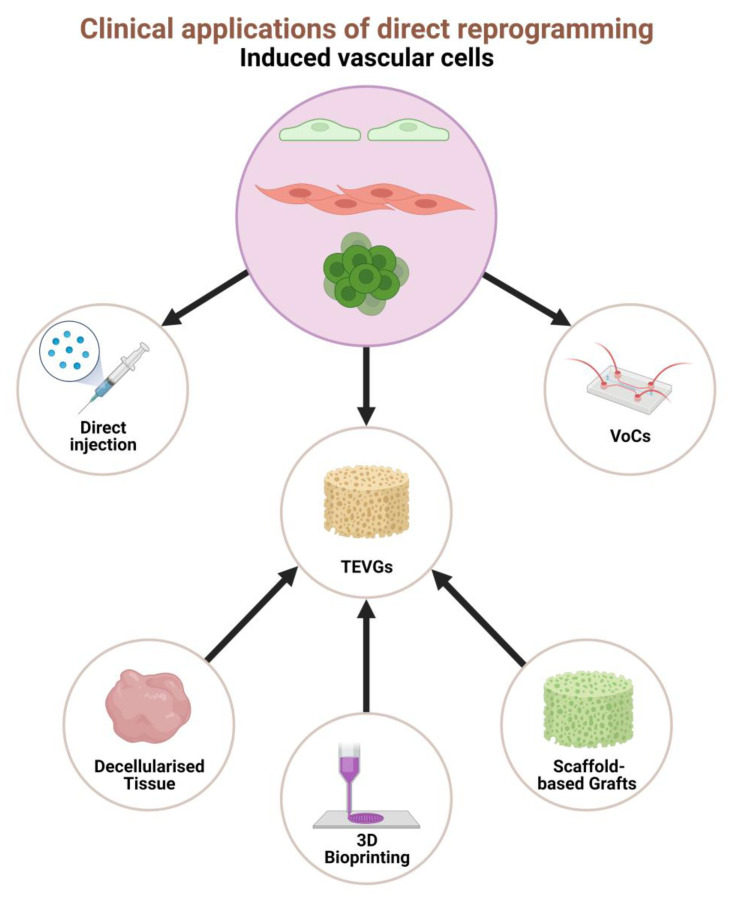
Spider diagram summarising the potential clinical applications of vascular cells generated through cellular reprogramming, such as cell therapy for vascular regeneration, tissue-engineered vascular grafts (TEVGs) for vascular grafting, and vascular on chips (VoCs) for disease modelling and high-throughput drug screening.

**Table 1 jfb-14-00021-t001:** Generation of endothelial cells through cellular reprogramming.

Reference	Source Cells	Transcription Factors	Culture Medium	Functional Outcome	Therapeutic Potential	Signalling Pathway	Limitations
Margariti et al., 2012 [10]	Human embryonic lung fibroblasts (HELF)	OCT4, SOX2, KLF4, C-MYC	EGM2	iECs were stable and formed patent vessels when constituted onto a decellularised vessel scaffold	Hindlimb ischaemia: increased capillary number and blood perfusion	SETSIP activation which promotes EC-specific gene expression	Embryonic cell source is ethically controversial
Li et al., 2013 [11]	Human neonatal fibroblasts	OCT4, KLF4	Differentiation medium II (50 ng/mL VEGF, 20 ng/mL bFGF, 0.1 mM 8-Br-CAMP)	Addition of 8-Br-cAMP increased transdifferentiation of fibroblasts into iECs	Murine hindlimb ischaemic model observed increased capillary number and blood perfusion	Not assessed	Conversion efficacy was low compared to studies using all 4 OSKM factors prior to sorting methods
Hong et al., 2017 [12]	Human umbilical artery smooth muscle cells	OCT4, SOX2, KLF4, C-MYC	EGM-2 (CC-3162) plus 25 ng/mL VEGF	SMCs are capable of trans-differentiation to iECs	Murine hindlimb ischaemic model observed increased capillary number and blood perfusion	OSKM upregulates VE-cadherin, HES4 and JAG1 which increases EC-specific gene expression	Lentiviral vectors possess safety risks. Viability of plasmid delivery confirmed but not explored
Chen et al., 2019 [13]	HELF	DKK3	EGM-2 plus EGM BulletKit (CC-3124) plus 10 ng/mL VEGF	iECs formed a patent monolayer in an ex vivo vascular graft	Formation of microvascular structures in vivo	Increases MET and VEGFR2, decreases miR-125a-5p and promotes Stat3	Embryonic cell source is ethically controversial
Ginsberg et al., 2012 [14]	Human amniotic fluid-derived cells	ETV2, FLI1, ERG1, TGFβ inhibition	EM (Medium 199, 15% FBS, 20 μg/mL endothelial cell supplement, 1X Pen/Strep, 20 units/mL Heparin)	iECs were generic and may hold potential for further subtype specification	In a regenerating mouse liver model, engraftment of iECs resulted in patent capillaries	Not assessed	Unsuccessfully with human postnatal cells. Use of amniotic cells is ethically controversial
Wong and Cooke, 2016 [15]	Human neonatal foreskin fibroblasts	ETV2, FLI1, GATA2, KLF4	EGM-2 plus 10 µmol/L SB341542	iECs uptake Ac-LDL and formed capillary-like networks	Not assessed	Not assessed	Therapeutic potential unknown
Kim et al., 2020 [16]	Human dermal fibroblasts	ETV2	EGM-2 plus doxycycline	iECs formed stable endothelial layers when seeded on a decellularised liver scaffold	Hindlimb ischaemia: improved angiogenic capabilities and blood perfusion	cAMP/EPAC/RAP1	iECs were not easily expandable
Morita et al., 2015 [17]	Human dermal fibroblasts	ETV2	EGM-2 medium (10 ng/mL recombinant human VEGF165, bFGF)	iECs displayed venous properties but adopted arteriole characteristics when combined with mural cells	Hindlimb ischaemia: improved angiogenic capabilities and blood perfusion	Modify DNA methylation states of EC genes	Extensive 50-day ETV2 exposure, lacking maturity, failed to induce NOS3
Lee et al., 2017 [18]	Human postnatal dermal fibroblasts	ETV2	EGM-2 plus DOX	Generation of early immature iECs, followed by matured iECs	Injection of early iECs into a murine hindlimb ischaemic model improved vessel generation and tissue perfusion	Not assessed	Early immature iECs failed to direct incorporation into host vasculature. Long timeline to cultivate mature iECs
Sayed et al., 2015 [9]	Human neonatal foreskin fibroblasts	Poly I:C (TLR3 agonist)	Maintenance medium (bFGF, VEGF, 0.1 mmol/L 8-Br-cAMP)	Innate immune activation is necessary for human fibroblasts to transdifferentiate into ECs effectively	Murine hindlimb ischaemic model observed increased expansion of host vasculature, blood perfusion and decreased tissue injury	Innate immune activation, TLR3/NF-κB/iNOS, epigenetic plasticity. Metabolic switching from oxidative phosphorylation to glycolysis	Low transdifferentiation efficacy. Therapeutic potential unknown, metabolic heterogeneity in iECs
McCoy et al., 2022 [19]	Human coronary artery smooth muscle cells (CASMCs)	miRNA	EGM-2 (2 µL/2 mL 8-Br-cAMP, 2 µL/mL SB 431542)	iECs exhibit high similarity to native ECs	Quicker limb reperfusion	Upregulation of NOTCH1, JAG1, and DLL4	Other miRNA targets need to be explored further

Ac-LDL, Acetylated-Low-Density Lipoprotein; bFGF, basic Fibroblast Growth Factor; 8-Br-CAMP, 8-Bromo-cAMP; cAMP, cyclic AMP; DKK3, Dickkopf WNT Signalling Pathway Inhibitor 3; DLL4, Delta Like Canonical Notch Ligand 4; DNA, Deoxyribonucleic Acid; DOX, Doxycycline; EC, Endothelial Cell; EGM-2, Endothelial Growth Medium-2; EM, Endothelial Medium; EPAC, Exchange Proteins directly Activated by cAMP; ERG1, Early Growth Response Protein 1; ETV2, ETS Variant Transcription Factor 2; FLI1, Friend Leukaemia Integration 1 Transcription Factor; GATA2, GATA-binding factor 2; HELF, Human Embryonic Lung Fibroblasts; HES4, Hes Family BHLH Transcription Factor; iEC, induced Endothelial Cell; iNOS, inducible Nitric Oxide Synthase; JAG1, Jagged Canonical Notch Ligand 1; MET, Mesenchymal-to-Epithelial Transition; miRNA, MicroRNA; miR-125a-5p, MicroRNA-125a-5p; NF-κB, Nuclear Factor Kappa B; NOTCH1, Neurogenic Locus Notch Homolog Protein 1; NOS3, Nitric Oxide Synthase 3; OSKM, OCT4 (Octamer-Binding Transcription Factor 4), SOX2 (SRY-Box Transcription Factor 2), KLF4 (Kruppel-Like Factor 4) and C-MYC (c-Myc proto-oncogene protein); Poly I:C, Polyinosinic-polycytidylic acid; RAP1, Ras-Related Protein 1; SB341542, TGF-β Receptor Kinase Inhibitor; SETSIP, Set Like Protein; Stat3, Signal transducer and activator of transcription 3; TGFβ, Transforming Growth Factor Beta; TLR3, Toll-Like Receptor 3; VEGF, Vascular Endothelial Growth Factor; VEGFR2, Vascular Endothelial Growth Factor Receptor 2.

**Table 2 jfb-14-00021-t002:** Generation of smooth muscle cells through cellular reprogramming.

Reference	Source Cell	Transcription Factors	Culture Medium	Functional Outcome	In Vivo Therapeutic Potential	Signalling Pathway	Limitations
Karamariti et al., 2013 [45]	HELF	OCT4, SOX2, KLF4, C-MYC	DM (MEM α, 10% FBS, 100 U/mL penicillin and streptomycin, 0.2 mM L-glutamine, 0.1 mM β-mercaptoethanol, 10 ng/mL PDGF-BB)	iVSMCs	Transplantation of iVSMCs-seeded decellularised vessel in mice increased survival	DKK3/Kremen1/Wnt signalling	Limited to HELF, Unknown efficacy of iVSMC generation, HELF is ethically controversial
Karamariti et al., 2018 [47]	HELF	DKK3	DMEM (ATCC, 10% EmbryoMax^®^ ES Cell Qualified FBS, 10 ng/mL LIF, 0.1 mM 2-mercaptoethanol) on a 0.04% gelatin substrate	VPCs, iVSMCs	Promotes stabilisation of atherosclerotic plaques by increasing SMCs and suppressing inflammation	DKK3/ATF6/TGFβ1	HELF is ethically controversial.
Hirai et al., 2018 [48]	MEF and adult dermal fibroblasts	Myocd, GATA6, MEF2C	SMC medium (DMEM/F-12, 10% KSR, 2 ng/mL recombinant human TGF-β1, 10 ng/mL human PDGF-BB, 1% penicillin-streptomycin)	iVSMCs	Not assessed	Not assessed	Partially reprogrammed iVSMCs

ATF6, Activating Transcription Factor 6; DKK3, Dickkopf WNT Signalling Pathway Inhibitor 3; DMEM, Dulbecco’s Modified Eagle Medium; FBS, Fetal Bovine Serum; GATA6, GATA-binding factor 6; HELF, Human Embryonic Lung Fibroblasts; iVSMCs, iPSC-derived Vascular Smooth Muscle Cells; KREMEN 1, Kringle Containing Transmembrane Protein 1; KSR, Knockout Serum Replacement; MEF, Mouse Embryonic Fibroblasts; MEF2C, Myocyte-Specific Enhancer Factor 2C; MEM α, Minimum Essential Medium α; Myocd, Myocardin; OSKM, OCT4 (Octamer-Binding Transcription Factor 4), SOX2 (SRY-Box Transcription Factor 2), KLF4 (Kruppel-Like Factor 4) and C-MYC (c-Myc proto-oncogene protein); PDGF-BB, Platelet-Derived Growth Factor-BB; SMC, Smooth Muscle Cell; TGFβ1, Transforming Growth Factor Beta 1; VPCs, Vascular Progenitor Cells.

**Table 3 jfb-14-00021-t003:** Generation of vascular progenitor cells through cellular reprogramming.

Reference	Source Cell	Transcription Factors	Culture Medium	Functional Outcome	In Vivo Therapeutic Potential	Signalling Pathway	Limitations
Kurian et al., 2013 [8]	Human neonatal and adult fibroblasts	OCT4, SOX2, KLF4, C-MYC	MIM (DMEM:F12,15 mg mL−1 stem cell–grade BSA, 17.5 μg mL^−1^ human insulin, 275 μg mL^−1^ human holo-transferrin, 20 ng mL^−1^ bFGF, 50 ng mL^−1^ human VEGF-165 aa, 25 ng mL^−1^ human BMP4, 450 μM monothioglycerol, 2.25 mM L-glutamine, 2.25 mM NEAA)	CD34^+^ angioblast-like bipotent progenitors	Forming functional blood vessels that integrated with host vasculature	Not investigated	Heterogenous cells
Zhang et al., 2017a [52]	Human adult and neonatal dermal fibroblast	OCT4, SOX2, KLF4, C-MYC	DMEM/F12 (20% KSR,10 ng mL−1 bFGF, 1 mM GlutaMAX, 0.1 mM NEAA, 55 μM β-mercaptoethanol)	Induced tripotent cardiac progenitor cells (iSMCs, iECs, iCMs)	Improved cardiac function and reduced adverse cardiac remodelling	Not investigated	Teratoma risk
Zhang et al., 2016 [53]	MEF	OCT4, SOX2, KLF4, C-MYC	ieCPC basal medium plus Advanced DMEM/F12: Neural basal (1:1) (1X N2, 1X B27 without Vitamin A, 1X Glutamax, 1X NEAA, 0.05% BSA, 0.1 mM β-ME) plus BACS, (5 ng/mL BMP4, 10 ng/mL Activin A, 3 μM CHIR99021, 2 μM SU5402)	BACS as a reliable prerequisite for the effective creation and ongoing renewal of ieCPCs	Directly produce CMs, ECs, and SMCs when exposed to the infarcted heart environment in vivo	Not investigated	Translational applicability of these cells
Pham et al., 2016 [54]	Human dermal fibroblasts	ETV2	Medium 200 (5% PRP, 5 ng/mL recombinant EGF, 1 ng/mL recombinant VEGF, 20 ng/mL insulin-like growth factor, 1 μg/mL ascorbic acid, 0.2 μg/ mL hydrocortisone, 22.5 μg/mL heparin, 1% antibiotic-antimycotic)	Unipotent iEPCs	Improve hindlimb ischemia	Not investigated	Venous not arterial ECs
Park et al., 2020 [55]	Mouse fibroblasts	ETV2, Fli1	VPC medium (10% FBS, 2 mmol/L L-glutamine, β-mercaptoethanol, penicillin/streptomycin, 10 ng/mL VEGF)	Self-renewal and biopotency iVPCs	Enhanced blood flow without tumour formation	Not investigated	Contamination of residual undifferentiated PSC

BACS, BMP4, Activin A, CHIR99021, SU5402; bFGF, basic Fibroblast Growth Factor; BMP4, Bone Morphogenetic Protein 4; BSA, Bovine Serum Albumin; CHIR99021, Glycogen Synthase Kinase 3 Inhibitor; CM, Cardiomyocyte; EC, Endothelial Cell; ETV2, ETS Variant Transcription Factor 2; Fli1, Friend leukaemia integration 1; iCM, induced Cardiomyocyte; iEC, induced Endothelial Cell; iEPC, induced Endothelial Progenitor Cell; iSMC, induced Smooth Muscle Cell; iVPC, induced Vascular Progenitor Cell; OSKM, OCT4 (Octamer-Binding Transcription Factor 4), SOX2 (SRY-Box Transcription Factor 2), KLF4 (Kruppel-Like Factor 4) and C-MYC (c-Myc proto-oncogene protein); MEF, Mouse Embryonic Fibroblasts; MEM, Minimal Essential Medium; MIM, Mesodermal Induction Medium; SMC, Smooth Muscle Cell; SU5402, Inhibitor of FGF, VEGF, and PDGF signaling; VEGF, Vascular Endothelial Growth Factor; VPC, Vascular Progenitor Cell.

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
