# Peer review of "Updated Perspectives on Direct Vascular Cellular Reprogramming and Their Potential Applications in Tissue Engineered Vascular Grafts"

_jfb, 2022, doi:10.3390/jfb14010021_

Round 1

Reviewer 1 Report

This is a well-written article on a very complex topic that is of interest to a wide scientific audience.

Please see the enclosed PDF for a point-by-point analysis.

Author Response

This is a well-written article on a very complex topic that is of interest to a wide scientific audience.

Response: Thank you very much for your positive comment about our review article and constructive suggestions to further improve the quality of our manuscript.

Please see the enclosed PDF for a point-by-point analysis.

Line-18: Please specify if this is a narrative review or systematic.

Response: Thank you, our manuscript is a narrative literature review, which has been specified in Abstract (Page 1, Abstract, line 7-8). 

Line-30: The introduction section is too long. Try to make it more concise and to the point.

Response: As suggested, we have shorten the introduction section to make it more concise (Page 2, paragraph 1 & 3). 

Line-101: The authors should state the originality of their approach.

Response: As suggested, following statement has now been included ‘To provide an in-depth knowledge of the current arterial direct reprogramming landscape and future research directions, we conducted the narrative literature review…’ (Page 2, last paragraph). 

Line-106: How was the search protocol performed?

Response: As suggested, we have now added the search protocol into the revised manuscript (Page 2, last paragraph). 

Line-907: Please delete.

Response: Deleted as suggested. 

Reviewer 2 Report

It is an interesting manuscript that provides a broad and exhaustive overview of recent updates on direct vascular cellular reprogramming, highlighting both the advantages and the limitations deriving from the application of this method.

Overall, the manuscript is well structured and deals with an emerging field by describing very innovative approaches.

However, I would suggest some changes to the authors in order to make it more readable and understandable.

 Here are my comments point by point:

In general, I suggest the authors review the English throughout the manuscript, there are some sentences that are difficult to understand (some specific ones will be indicated below).

Furthermore, since this is a literature review, I suggest that the authors insert a short paragraph (after the introduction section) reporting the keywords used in the research, the databases used for this purpose and any inclusion/exclusion criteria.

 Introduction

-     Lines 38 and 39: I recommend improving the English in order to clarify the concept: rupture and not rapture!

-          Line 47: please insert a reference regarding he limitations of the revascularization interventions.

-          Lines 85-91:  please clarify better the concept and improve the English.

-          Lines 96 and 100: please insert a reference describing the third approach inducing a progenitor-like state.

 Endothelial Cell Generation

-          Line 115: in round brackets reverse the order by placing Table 1 which is the first to appear and then Figure 1.

-          Table 1: I suggest to graphically improve the table and I suggest inserting the reference number next to the name of the first author (for example Margariti et al. is [14]). I also suggest inserting a legend with the description of the abbreviations so that it can help the reader who, in this way, will not have to look for the abbreviations in the text. Other minor points:

o   second line of the table (Li et al., 2013): the description of “culture media composition is lacking a round bracket at the end.

o   sixth line of the table (Ginsberg et al., 2012): the description of “culture media composition is lacking a round bracket at the end.

-      Figure 1: In the figure I understand that there is a color code for each topic covered, I suggest the authors write in the legend what it refers to, for example: Pink: Pluripotency factor-based reprogramming; ...Green: Lineage-specific transcription factors- the advent of ETV2....Furthermore, since they are not always fibroblasts but also other cell types, I would leave only somatic cell at the top without "(e.g. fibroblasts)".

Pluripotency factor-based reprogramming

-          Line 170: please correct the word “lowed” if it was intended as lowered.

-          Lines 244-245: please clarify the concept, by explicating to what kind of proteins or factors modified mRNA is referring to.

-          Lines 334-342:  please clarify the concept of “epigenetic plasticity”, which appeared to be a key effect of the agonist TLR3, allowing fibroblast to differentiate in iECs. 

-          Lines 351-367: please give more details about the cited study by mentioning the study model used.

-          Lines 494-502: please give more details about the cited study by mentioning the study model used, specifically which type of cells were induced to differentiate.

Smooth Muscle generation:

-          Line 405: in round brackets reverse the order by placing Table 2 which is the first to appear and then Figure 2. Also, the acronym HELF was first written in Table 1, so I suggest writing the full word where it is first named.

-          Table 2 and Figure 2: I ask the authors to follow the same suggestions for Table 1 and Figure 1.

Vascular Progenitor Cells:

-          Line 473: in round brackets reverse the order by placing Table 3 which should be before figure 3 to respect the order followed for the other tables and figures.

-          Table 3 and Figure 3: I ask the authors to follow the same suggestions for Table 1 and Figure 1.

Tissue engineered vascular graft

-          Line 586: I suggest correcting the word “SUSTAINABLY” if it was intended as SUSTAINABILITY. Also suggests adding an explanatory reference to the sentence.

-          Figure 4: I suggest improving graphically, in particular to move the main text in bold black outside the graphic.

Decellularised tissue

-          Lines 650-652: please clarify the concept and improve the English.  

Bioprinting

-          Line 658: please correct as “5.2. 3D Bioprinting”

-          Line 660:  please add a reference study regarding the 3D bioprinting.

-          Lines 667-670: please clarify the concept and improve the English.

The ethics of direct reprogramming                      

-          Lines 720-724: please clarify the concept and improve the English.

       Current challenges e future perspectives

-          Line 745: Please add a reference about the trial-and-error method.

-          Line 859: Please correct as “7.6. 2D in vitro….”

Conclusion

-          Line 887: Please add references about the preclinical studies.

Author Response

It is an interesting manuscript that provides a broad and exhaustive overview of recent updates on direct vascular cellular reprogramming, highlighting both the advantages and the limitations deriving from the application of this method.

Overall, the manuscript is well structured and deals with an emerging field by describing very innovative approaches.

However, I would suggest some changes to the authors in order to make it more readable and understandable.

Response: Thank you very much for your nice comments about our work. Your constructive suggestions to further improve the quality of our manuscript were hugely appreciated

Here are my comments point by point:

In general, I suggest the authors review the English throughout the manuscript, there are some sentences that are difficult to understand (some specific ones will be indicated below).

Response: Thanks a lot for your critical reading and pointing out all the specific examples for us to consider for further improvement. As suggested, we have now carefully proofread the manuscript to avoid possible grammatical errors. Our revised manuscript has also been proofread by an English native speaker.

Furthermore, since this is a literature review, I suggest that the authors insert a short paragraph (after the introduction section) reporting the keywords used in the research, the databases used for this purpose and any inclusion/exclusion criteria.

Response: As suggested, we have now added the search protocol into the revised manuscript (Page 2, last paragraph).

Introduction

-Lines 38 and 39: I recommend improving the English in order to clarify the concept: rupture and not rapture!

Response: As suggested by Reviewer-1, we have shorten the introduction section to make it more concise (Page 2, paragraph 1 & 3).

-Line 47: please insert a reference regarding he limitations of the revascularization interventions.

Response: Added as suggested (Ref-2 & 3).

- Lines 85-91:  please clarify better the concept and improve the English.

Response: As suggested, we have rephrased this paragraph to better present the research concept (Page 2, paragraph 3).

- Lines 96 and 100: please insert a reference describing the third approach inducing a progenitor-like state.

Response: Added as suggested (Ref-8).

Endothelial Cell Generation

- Line 115: in round brackets reverse the order by placing Table 1 which is the first to appear and then Figure 1.

Response: Modified as suggested (page-3).

- Table 1: I suggest to graphically improve the table and I suggest inserting the reference number next to the name of the first author (for example Margariti et al. is [14]). I also suggest inserting a legend with the description of the abbreviations so that it can help the reader who, in this way, will not have to look for the abbreviations in the text. Other minor points:

o   second line of the table (Li et al., 2013): the description of “culture media composition is lacking a round bracket at the end.

o   sixth line of the table (Ginsberg et al., 2012): the description of “culture media composition is lacking a round bracket at the end.

Response: As suggested, the reference numbers for each paper have now been inserted into Table 1, and all the abbreviations have now been added into the table legend (pages 3-5, Table 1). Moreover, the missed round bracket have now been added into its respective position.

- Figure 1: In the figure I understand that there is a color code for each topic covered, I suggest the authors write in the legend what it refers to, for example: Pink: Pluripotency factor-based reprogramming; ...Green: Lineage-specific transcription factors- the advent of ETV2....Furthermore, since they are not always fibroblasts but also other cell types, I would leave only somatic cell at the top without "(e.g. fibroblasts)".

Response: We have now added the color code and all the abbreviations into the figure legend, as well as modified the figure as suggested (Page 6, Figure-1).

Pluripotency factor-based reprogramming

- Line 170: please correct the word “lowed” if it was intended as lowered.

Response: Corrected as suggested (Page 7, last paragraph).

- Lines 244-245: please clarify the concept, by explicating to what kind of proteins or factors modified mRNA is referring to.

Response: As suggested, following statement ‘However, a study used modified mRNA encoding ETV2, FLI1, GATA2, and KLF4 to…’ has now been added into the revised manuscript (Page 9, 2nd paragraph, line 7).

- Lines 334-342: please clarify the concept of “epigenetic plasticity”, which appeared to be a key effect of the agonist TLR3, allowing fibroblast to differentiate in iECs.

Response: As suggested, following statement ‘In their introductory study, an agonist of TLR3, polyinosinic:polycytidylic acid (polyI:C; PIC), induced a state of epigenetic plasticity, namely a global changes in epigenetic modifiers that increase the probability for an open chromatin state, in human fibroblasts…’ has now been added into the revised manuscript (Page 11, 1st paragraph, line 2-3).

- Lines 351-367: please give more details about the cited study by mentioning the study model used.

Response: Added as suggested (Page 11, 3rd paragraph, line 2-6).

- Lines 494-502: please give more details about the cited study by mentioning the study model used, specifically which type of cells were induced to differentiate.

Response: We have mentioned that ‘human adult dermal fibroblasts’ were used as the starting cells (Page 19, 3rd paragraph, line 1).

Smooth Muscle generation:

- Line 405: in round brackets reverse the order by placing Table 2 which is the first to appear and then Figure 2. Also, the acronym HELF was first written in Table 1, so I suggest writing the full word where it is first named.

Response: We have reversed the order by placing Table 2 first (Page 12, 3rd paragraph, line 3). The full name of HELF was given in Table 1.

- Table 2 and Figure 2: I ask the authors to follow the same suggestions for Table 1 and Figure 1.

Response: Thank you! As suggested, we have now modified all the required changes in Table 2 and Figure 2 (Page 12-15; Table 2; Figure 2).

Vascular Progenitor Cells:

- Line 473: in round brackets reverse the order by placing Table 3 which should be before figure 3 to respect the order followed for the other tables and figures.

Response: We have reversed the order by placing Table 3 first (Page 15, last line). The full name of HELF was given in Table 1.

- Table 3 and Figure 3: I ask the authors to follow the same suggestions for Table 1 and Figure 1.

Response: Thank you! As suggested, we have now modified all the required changes in Table 3 and Figure 3 (Page 16-19; Table 3; Figure 3).

Tissue engineered vascular graft

- Line 586: I suggest correcting the word “SUSTAINABLY” if it was intended as SUSTAINABILITY. Also suggests adding an explanatory reference to the sentence.

Response: Collected as suggested (page 21, 1st paragraph, lines 11-12). Also, two references have now been added to support the statement (Reference- 60 & 61).

- Figure 4: I suggest improving graphically, in particular to move the main text in bold black outside the graphic.

Response: Thank you! Modified as suggested (Figure 4).

Decellularised tissue

- Lines 650-652: please clarify the concept and improve the English.

Response: This sentence has now been removed from the revised manuscript.

Bioprinting

- Line 658: please correct as “5.2. 3D Bioprinting”

Response: Corrected as suggested (Page 23).

- Line 660:  please add a reference study regarding the 3D bioprinting.

Response: As suggested, one reference has now been added to support our statement (reference-75).

- Lines 667-670: please clarify the concept and improve the English.

Response: The mentioned contents have now been removed from the revised manuscript since they are not very relevant to this review.

The ethics of direct reprogramming                     

- Lines 720-724: please clarify the concept and improve the English.

Response: The entire paragraph has now been removed from the revised manuscript as it is just a repeat of the statement in above sections.

Current challenges e future perspectives

- Line 745: Please add a reference about the trial-and-error method.

Response: As suggested, one reference has now been added to support our statement (reference-34).

- Line 859: Please correct as “7.6. 2D in vitro….”

Response: Corrected as suggested (Page-27).

Conclusion

- Line 887: Please add references about the preclinical studies.

Response: Sorry for the possible misleading. Here we didn’t discuss any specific study (or data set), instead we tried to summarise the data from the studies discussed in the entire review. Accordingly, we have modified the sentence to make it clear (‘Data from the preclinical studies discussed in this review have confirmed the potential for significant therapeutic bene-fits…’) (Page-27, last paragraph, lines 5-6).